# Context-Based Patterns in Machine Learning Bias and Fairness Metrics: A Sensitive Attributes-Based Approach

**Tiago P. Pagano** [1], **Rafael B. Loureiro** [1], **Fernanda V. N. Lisboa** [2], **Gustavo O. R. Cruz** [2], **Rodrigo M. Peixoto** [3], **Guilherme A. de Sousa Guimarães** [3], **Ewerton L. S. Oliveira** [4], **Ingrid Winkler** [5] and **Erick G. Sperandio Nascimento** [1,6,*]

1. Computational Modeling Department, SENAI CIMATEC University Center, Salvador 41650-010, BA, Brazil
2. Computer Engineering, SENAI CIMATEC University Center, Salvador 41650-010, BA, Brazil
3. Software Development Department, Salvador 41650-010, BA, Brazil
4. HP Inc. Brazil R&D, Porto Alegre 90619-900, RS, Brazil
5. Management and Industrial Technology Department, SENAI CIMATEC University Center, Salvador 41650-010, BA, Brazil
6. Surrey Institute for People-Centred AI, School of Computer Science and Electronic Engineering, Faculty of Engineering and Physical Sciences, University of Surrey, Guildford GU2 7XH, UK
* Correspondence: erick.sperandio@surrey.ac.uk

**Abstract:** The majority of current approaches for bias and fairness identification or mitigation in machine learning models are applications for a particular issue that fails to account for the connection between the application context and its associated sensitive attributes, which contributes to the recognition of consistent patterns in the application of bias and fairness metrics. This can be used to drive the development of future models, with the sensitive attribute acting as a connecting element to these metrics. Hence, this study aims to analyze patterns in several metrics for identifying bias and fairness, applying the gender-sensitive attribute as a case study, for three different areas of applications in machine learning models: computer vision, natural language processing, and recommendation systems. The gender attribute case study has been used in computer vision, natural language processing, and recommendation systems. The method entailed creating use cases for facial recognition in the FairFace dataset, message toxicity in the Jigsaw dataset, and movie recommendations in the MovieLens100K dataset, then developing models based on the VGG19, BERT, and Wide Deep architectures and evaluating them using the accuracy, precision, recall, and F1-score classification metrics, as well as assessing their outcomes using fourteen fairness metrics. Certain metrics disclosed bias and fairness, while others did not, revealing a consistent pattern for the same sensitive attribute across different application domains, and similarities for the statistical parity, PPR disparity, and error disparity metrics across domains, indicating fairness related to the studied sensitive attribute. Some attributes, on the other hand, did not follow this pattern. As a result, we conclude that the sensitive attribute may play a crucial role in defining the fairness metrics for a specific context.

**Keywords:** bias; fairness; sensitive attribute; machine learning; artificial intelligence





## 1. Introduction

Prediction-based decision algorithms are widely used in industry and are rapidly gaining traction with governments and organizations. These techniques are already widely used in lending, contracting, and online advertising, and they are becoming more common in fields such as public health, immigration detention, and criminal pre-trial systems [1]. As machine learning (ML) becomes more common in decision-making applications, systems that affect people's lives must address ethical concerns to ensure that decisions are made fairly and objectively [2].

Numerous studies on bias and fairness have been conducted, taking into account the constraints imposed by legal restrictions, corporate procedures, societal conventions, and ethical standards [2]. Fairness can be defined as a social idea of value judgment and, therefore, a subjective concept that varies across cultures, nations and institutions. Bias, on the other hand, is a systematic error that modifies human behaviors or judgments about others due to their belonging to a group defined by distinguishing features, such as gender or age [3].

Identifying and improving bias and fairness is a difficult task since they are perceived differently in different societies. As a result, their criteria take into account the individual's history, as well as cultural, social, historical, political, legal, and ethical concerns [4]. New data science, artificial intelligence, and machine learning approaches are required to analyze model performance for sensitive social variables such as race, gender, and age [1].

Furthermore, the predicament is exacerbated if the primary technological applications lack machine learning models concerned with the explainability of decisions made, or if these models can only be analyzed by the team that created them, limiting the information that can be inferred from these models [5]. One method is to comprehend the decisions without having to comprehend every step taken by the algorithm [6].

Data from the scenario in which the model will be used, and data about the model itself are used to describe the situation and provide context, that is, any information that can be used to describe the status of an entity [7]. Individuality, action, location, time, and relationships are the five categories into which the context information elements fall. Whereas location and time are primarily responsible for establishing interactions between entities and allowing context information to flow between entities, activity is primarily responsible for determining the relevance of context components in specific situations. The model, data, and fairness criteria are entities, and their interactions are the subject of this study.

Because they are context-dependent, determining the best metric for a problem is still an open question in the research. Different contexts require different approaches, and understanding the metric is critical for effective results. For instance, Anahideh et al. [8] attempted to identify a set of ideal metrics for the context based on the sensitive attributes.

Many current approaches to bias and fairness are applications for a specific problem [9–12]. There are several approaches for identifying bias and fairness, known as fairness metrics, and the diversity makes determining the best evaluation criteria for a problem difficult [13]. Some solutions provide tools to help developers identify bias and fairness, such as AIF360 [14], FairLearn [15], Tensorflow Responsible AI [4,16,17] and the Aequitas [18]. However, many of these approaches fail to account for the relationship between the application context and the sensitive attributes associated with it, allowing ineffective fairness metrics to be implemented [8].

In the domain of computer vision (CV), for example, it is difficult to identify and separate the vast amount of visual information in the environment. Machines can categorize things, animals, and humans using algorithms, optical and acoustic sensors, and other tools [19]. However, due to the numerous sources of bias that may arise during model training and evaluation, these machines may struggle to distinguish between faces, skin tones, and races [20]. This usually happens when the context is ignored during model development, such as when underrepresented user demographics are not taken into account in the training data [21].

Similarly, natural language processing (NLP) applications, such as language translation [22] and the automatic removal of offensive comments [23], are critical for systems that interact with people. Transformer architectures have been employed to extract implicit meanings from vast amounts of text [24]; nevertheless, a major concern with such models is the negative generalization of terms that should not have a negative connotation in a broad context, such as *gay* or *woman* [25,26].

Recommendation systems (RS) are widely used in everyday life, such as catalog streaming systems and online retailer user perception of product orders. These programs

are classified as rankers because their function is to determine the current preferences in the input and output lists of suggestions [27]. To evaluate a specific product or service, their learning approach requires qualitative interactions between users and products, such as the likes and dislikes system. Amazon's catalog recommendation systems generate consumer profiles and offer related products based on their preferences [28]. Another recommendation system's feature is the ability to connect similar profiles, recognizing that if one user has rated a product positively, another similar user is likely to do the same. Such systems, however, rely on massive amounts of historical data, which may contain unrealistic training samples or reflect historical inequalities [29]. Furthermore, biased systems that favor specific groups may create vicious cycles for recommendations, reinforcing negative biases.

Hence, due to the difficulties in analyzing and determining which metrics of bias and fairness for machine learning models are more suitable than others for different application domains for a specific sensitive attribute, this study aims to methodologically analyze patterns in several metrics for identifying bias and fairness, applying the gender-sensitive attribute as a case study for three different areas of applications in machine learning models: computer vision, natural language processing, and recommendation systems.

The contributions of this work include:

- Identification of the fairness metrics that have similar behavior in different models for the same sensitive attribute;
- Determination of the sensitive attribute and the fairness metric as an element for context definition;
- Verification whether different use cases with the same sensitive attribute have similar contexts.

This work is organized as follows: Section 2 describes the research method and presents the domains and models that were employed, Section 3 presents the results of the models' bias and fairness analyses for the three problems, and Section 4 provides the final considerations and suggestions for future research.

## 2. Materials and Methods

We divided the method into four steps, as shown in Figure 1. Step (1) is to specify and analyze datasets in tasks for computer vision, natural language processing, and recommendation systems as well as identifying key components for model development. Step (2) involves the development of models using traditional machine learning architectures, which leads to the acquisition of research objects for fairness analysis. Step (3) entails calculating the models' fairness metrics for the gender attribute using the materials and methods developed. Finally, Step (4) is to evaluate and analyze the obtained results in order to discover relevant links between the model and the fairness results, thereby contributing to the context definition.

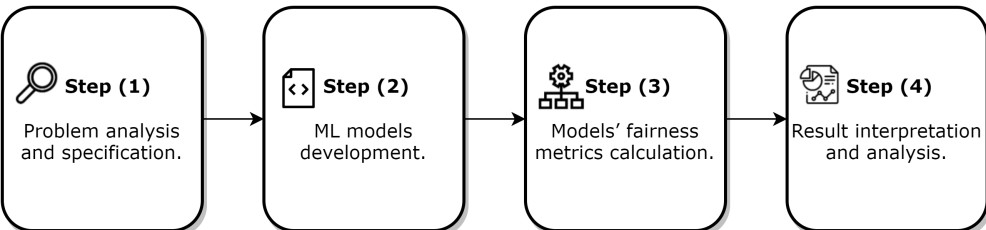

**Figure 1.** Research methodology flow chart showing the steps to recognize the patterns for bias and fairness through a sensitive attribute.

The generated models were evaluated using the classification metrics accuracy, precision, recall, F1-score and AUPRC. Each of these classification metrics is defined below:

- Accuracy: a metric that computes the percentage of correct predictions made by the model in relation to the total number of predictions. This metric should be

used when the dataset's class distribution is even and avoided when the dataset is imbalanced [30].

$$Accuracy = \frac{TP + TN}{TP + TN + FP + FN} \quad (1)$$

- Precision: a metric that quantifies the positive predictions made by the model were actually positive, by dividing the total true positives by the sum of true positives and false positives [31].

$$Precision = \frac{TP}{TP + FP} \quad (2)$$

- Recall: a metric that calculates how many true positives from the dataset were correctly identified by the model by dividing the total number of true positives by the sum of true positives and false negatives [31].

$$Recall = \frac{TP}{TP + FN} \quad (3)$$

- F1-score: a metric defined as the harmonic mean of precision and recall, so that both have the same weight for model evaluation. This indicates that the higher the value of the F1-score, the higher the score of the other two metrics. This metric can be used for imbalanced datasets and is more reliable for indicating model quality [31].

$$\text{F1-score} = \frac{2 * (Recall * Precision)}{(Recall + Precision)} \quad (4)$$

- AUPRC: the area under the precision-recall curve (AUPRC) metric is used to evaluate models for imbalanced datasets and are defined as the fraction of positive [32], where this fraction is determined by the ratio of positive (P) and negative (N) according to Equation (5) [33].

$$AUPRC = \frac{P}{P + N} \quad (5)$$

For each classification class (0 or 1), we calculated the precision, recall, and F1-score and present their macro average (macro avg) and weighted average (weighted avg). Macro avg computes the unweighted average metrics for the labels without taking into account label imbalance. Weighted avg, on the other hand, computes the weighted average based on the number of true instances for each label.

*2.1. Datasets*

We utilized Face Recognition–FairFace Challenge (FIFC), Jigsaw Unintended Bias in Toxicity Classification (JUBTC) and MovieLens100K datasets for the CV, NLP and RS problems, respectively [21].

The FIFC is a challenge organized by ChaLearn that aims to encourage research on fair face recognition and provide a new dataset with accurate annotations, with competition participants required to deliver solutions that improve bias and fairness regarding sensitive dataset attributes [34].

The IARPA Janus Benchmark-C (IJB-C) dataset was used for the FIFC, and it contains 31,334 still images (21,294 with faces and 10,040 without faces) and 117,542 frames from 11,779 full-motion videos, with an average of 33 frames per person. The IJB-C dataset's contributions are linked to facial recognition and biometric features that include human faces in various positions, as well as using people with distinct professions to avoid the problem of using celebrities. People whose physical appearance is directly related to their professions, such as actors and models, may not be representative of the global population [35]. The IJB-C received an addition of 12,549 public domain images, as well as a re-annotation, to achieve the contest's goal. However, it still remains imbalanced, with



more data belonging to bright skin and male gender than dark skin and female gender, which may produce a bias related to both skin color and gender attributes.

About JUBTC, in 2017, during the first toxicity classification challenge, it was discovered that the models incorrectly associated gender identities with toxicity, as these are frequently used with toxic connotations in the various bases that comprised the dataset of the challenge. Training a model with these data causes the network's learning to reproduce these actual biases [25]. The JUBTC contest was established in 2019 to address this issue, created by the Conversation AI team and hosted on the Kaggle platform, with the goal of encouraging the development of models that recognize toxicity and reduce this type of unintended bias using the Civil Comments Dataset (CCD) [36]. The CCD is a collection of data extracted from the Civil social network [25].

The dataset in question has 45 columns and 1.9 million examples. Its main columns are target, identity-attack, insult, obscene, sexually explicit, and threat. They determine the overall toxicity as well as the intensity of that toxicity for each comment. The dataset also includes attributes for gender identities, ethnicity and race, religion, and physical or psychological disabilities.

Except for comment-text, which is a string of text, and columns that refer to the date, ID, or the feelings they triggered in the annotation team, the attributes are mostly of the decimal type, ranging between 0 and 1.

While detecting toxicity is difficult, the most difficult challenge in JUBTC is detecting it without introducing bias or removing fairness from the model. This bias and fairness manifest themselves in the model's association between a frequently attacked identity and the labeling of a comment as toxic. This is due to the negative treatment of certain identities, such as the phrase *I am a gay woman*, which is considered toxic by several models due to the presence of the word *gay*, which is frequently used in prejudiced comments.

The MovieLens dataset was collected by GroupLens [37], a research group from the University of Minnesota, and includes a collection of movie ratings from the MovieLens website, which is a movie recommendation service. There are five versions included, with different sizes and attributes in each one of them. The dataset version used is "100k", which collected 100,000 reviews from 1000 users between 1997 and 1998 for 1700 films. In addition to information about the films and ratings, the reviews include gender, age, occupation, and postcode information. Users with fewer than 20 ratings or who did not provide complete demographic information had their data erased. It is noted that user attributes may cause or reproduce a bias since the data classes are not balanced, such as in gender and age. Additionally, some users may have more film recommendations than others with similar behavior but different classes.

### 2.2. Models

We utilized Visual Geometry Group (VGG19), Bidirectional Encoder Representations from Transformers (BERT) and Wide and Deep models for FIFC, JUBTC and MovieLens100K datasets, respectively.

We used the VGG19 architecture to build a model using the FIFC. The identification of bias and fairness in models is independent of their performance in classification and regression metrics, such as accuracy, error, and so on. For example, there could be a model with a high hit rate and low error that produces unfair results.

The VGG19 architecture, designed for large-scale image classification, went through a transfer learning process with ImageNet dataset weights, adding the following layers in the following order: Flatten, Dense (64 neurons), Batch Normalization, Dropout, Dense (10 neurons), Batch Normalization, Dense (6 neurons), and Dense (1 neuron). The ReLU activation function was present in all dense layers. The target was the head position attribute (HEAD_POSE), and the input was 240 by 240 pixels in BGR format, with the activation function Sigmoid, the optimizer Adam, and the loss function Binary Cross-Entropy.

The BERT base uncased is used in the model developed for JUBTC. Token generation with its own tokenizer of size 200. It was set up for binary classification, predicting the toxicity intensity in comments ranging from 0 to 1.

The Wide and Deep is divided into two parts, the Wide and the Deep part. The Wide part is used to represent the co-occurrence of features in the learning sample and contains fundamental and traversal characteristics as input attributes, such as a direct link between the user and the movie, with the user acting as the outcome [38].

The Deep part employs a feedforward neural network to classify the input, which is frequently a string such as a user identity. First, from these high-dimensional classification characteristics, a low-dimensional float vector with a randomly initialized value is created [38].

*2.3. Fairness Metrics*

Step (3) defines the identification of bias and fairness in the developed models.

There are some metrics defined in the literature to identify fairness in machine learning, such as statistical parity value (SPV), equalized odds (EO), predictive equality (PE), predictive parity (PPV), average odd difference (AOD), equal of opportunity (EOO), fnr difference (FNRDif), false negative rate disparity (FNRD), predicted positive ratio disparity (PPRD), false positive rate disparity (FPRD), error disparity (ED), true positive rate disparity (TPRD), and AUPRC, among others [21].

Before discussing the goals of each fairness indicator, it is critical to understand the values of true positive (TP), true negative (TN), false positive (FP), and false negative (FN). Positive values are those obtained by the model that are consistent with its objectives. For instance, a model's prediction that a message is poisonous will be regarded as positive if the message proves to be harmful. If the message is not harmful, the opposite is perceived negatively. As a result, the amount of TP denotes positive values correctly predicted by the model, whereas the amount of FP denotes positive values incorrectly predicted by the model. TN denotes the negative values that the model correctly predicted, whereas FN denotes the negative values that the model incorrectly predicted.

Because most users have not interacted with the movies in the catalog, establishing ground-truth values is difficult; therefore, the values of TP, FP, TN, and FN are acquired differently for RS. Only movie suggestions that the user had previously interacted with were considered for model evaluation in order to create a confusion matrix of the model recommendations. We used a technique as a foundation, where TP values denote a film that has been endorsed by both the user and the system. TN values, on the other hand, denote movies that are not user-approved and are not highly recommended [39].

The TP, FP, TN, and FN values were used to calculate the following fairness metrics, which are categorized as follows:

(a)     Disparity and parity metrics: calculated based on the ratio of one measure for a benefited group to a non-benefited. If the ratio is 1, the groups are equal.

- FNR disparity: disparities calculated based on FNR.
- PPR disparity: disparities calculated based on predicted positive ratio $PPR = \frac{PP}{PP+PN}$.
- FPR disparity: disparities calculated based on false positive rate $FPR = \frac{FP}{FP+TN}$. FPR is the proportion of cases with incorrectly detected negative conditions as positive. In the example, it is the rate of non-toxic messages mistakenly predicted as toxic.
- Error disparity: disparities calculated based on each group's error, defined by $Error = \frac{FP+FN}{Total}$.
- TPR disparity: disparities calculated based on true positive rate $TPR = \frac{TP}{TP+FN}$. TPR is the proportion of positive cases correctly detected. In the example, it is the rate of toxic messages correctly predicted as toxic.

- Predictive parity: this metric is satisfied by a classifier if the $PPV = \frac{TP}{TP+FP}$ of both the protected and unprotected groups is equal to the probability that a subject with positive predictive value truly belongs to the positive class. In the example, it implies that the toxicity classification of skin-color-related messages has the same probability for distinct classes to be classified as toxic.
- Statistical parity: both toxic and non-toxic messages should have equal chances to occur. Both methods should have the same chance of making a positive prediction $(TP + FP)$.

(b) Difference metrics: Calculated based on the difference between the benefited group to a non-benefited. If the result is 0, the groups have the same value.

- FOR difference (FORD): the difference between the false omission rate $FOR = \frac{FN}{TN+FN}$ of the benefited group and the other groups. The FOR is the fraction of incorrectly predicted positive cases out of all predicted negative cases. In the example, it corresponds to the rate of messages incorrectly classified as non-toxic for a total of non-toxic messages.
- FNR difference: the difference between the false negative rate $FNR = \frac{FN}{TP+FN}$ of the benefited group and the other groups. The FNR is the probability of a positive case not being detected. In this example, it is the proportion of toxic messages misclassified as non-toxic.
- Average odds difference (AOD): this metric is defined by Equation $AOD = \frac{1}{2} * \left( \frac{FP_0}{FP_0+TN_0} - \frac{FP_1}{FP_1+TN_1} + \frac{TP_0}{TP_0+FN_0} - \frac{TP_1}{TP_1+FN_1} \right)$, indicating the difference between the values correctly classified as positive and negative, of benefited versus the other group (0 and 1).

(c) Other Fairness metrics.

- AUPRC: summarizes the precision–recall curve as the weighted average precision of each threshold, with an increasing recall of the previous threshold used as weight, following the equation: $AUPRC = \sum_n (R_n - R_{n-1})P_n$, where $P_n$ and $R_n$ are, respectively, the precision and recall at threshold 'n'.
- Equalized odds: toxic and non-toxic messages should have the same false alarm rate, i.e., toxic messages should be predicted as non-toxic and vice versa. As a result, the false positive $FPR = \frac{FP}{TN+FP}$ and false negative $FNR = \frac{FN}{TP+FN}$ rates for toxic and non-toxic messages should be the same. Toxic messages should be equally likely to occur.
- Predictive equality: this metric is satisfied by a classifier if both the benefited and non-benefited groups have equal $FPR = \frac{FP}{TN+FP}$, i.e., the probability of a subject in the negative class having a positive predictive value. In our example, this means that the probability of a true toxic message being misclassified as non-toxic must be equal for all classes of a given sensitive attribute.
- Equal of opportunity: this metric is satisfied by a classifier if both the protected and unprotected groups have equal $TPR = \frac{TP}{TP+FN}$, requiring non-discrimination only in the favoured outcome group. The example implies that, regardless of classification, only toxic messages (TP + FN) are considered for the toxicity classification of messages referencing skin color.

Bias and fairness can be detected by assessing the models' fairness metrics. To calculate the fairness metrics, it is necessary to first define a benefited group. In addition, the two most representative groups are chosen based on the proportion of total group occurrences to the total records in the dataset. After selecting the two most representative groups in quantitative terms in the dataset, the proportion of both is calculated according to Equation (6), and the one with the higher value is considered the benefited class. Equation (6) is used before the fairness metrics calculation to define the benefited group. Once the group is defined, we calculate the fairness metrics:

$$Representativeness_{class} = \frac{FNR}{FPR} \tag{6}$$

The fairness metrics should be within 20% of the benefiting class [18].

The fairness metrics obtained from each of the models VGG19, BERT, and Wide and Deep were used to analyze the similarities in the results. If all classes of their sensitive attributes are within the threshold for a given metric, the model is considered fair; otherwise, it is considered unfair. We compare the models for each metric, determining whether the same metric produces the same fairness result (fair or unfair) across models. If the results are the same, the metric is considered representative in assessing bias and fairness for the sensitive attribute. To assist in model comparison, we categorize each metric as similar if the metric produces the same results across models, and different otherwise.

Figure 2 illustrates a diagram of the method for analyzing the three models' fairness metrics.

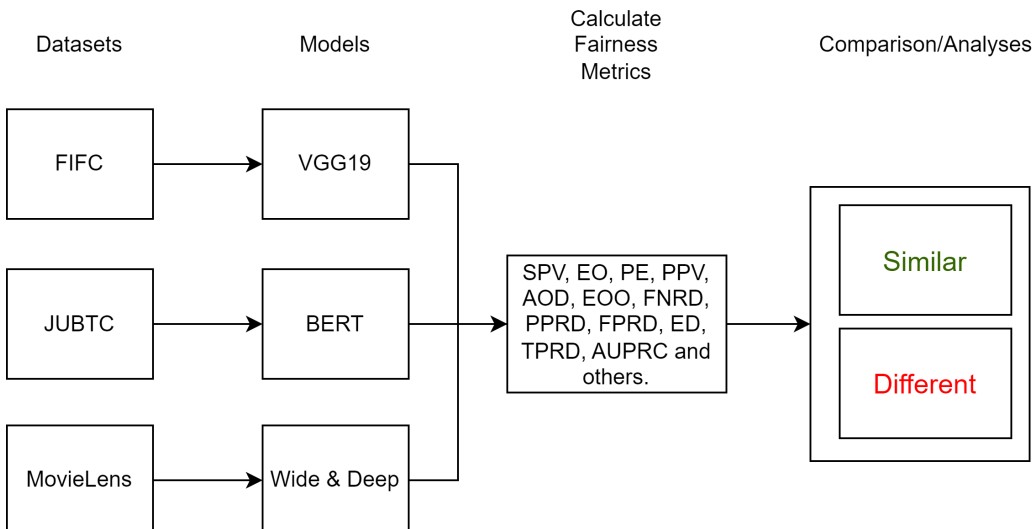

**Figure 2.** Method for comparing fairness metric similarities.

### 3. Results

*3.1. Analysis of the Developed Model for the FIFC Problem*

In Step (1), we analyzed that the FIFC dataset contained the following sensitive attributes: gender (male and female), skin color (light corresponding to Fitzpatrick scale I–III, dark corresponding to Fitzpatrick scale IV–VI), and five legitimate attributes, including age (0–34, 35–64, 65+), head position (front, other), image category (still or video image), use or lack of glasses, and face cutout size. However, some characteristics, such as gender and race or ethnicity, are imbalanced.

We also changed the structure of the challenge to resemble a classification problem to identify the head position (HEAD_POSE), so that the value 0 represents the frontal position and the value 1 represents another position of the individual. Figure 3 depicts that the HEAD_POSE attribute distribution is imbalanced. Table 1 displays values for the sensitive attribute gender.

**Table 1.** Gender categories for FIFC.

| Gender | Label |
|:------:|:-----:|
| Male | 0 |
| Female | 1 |

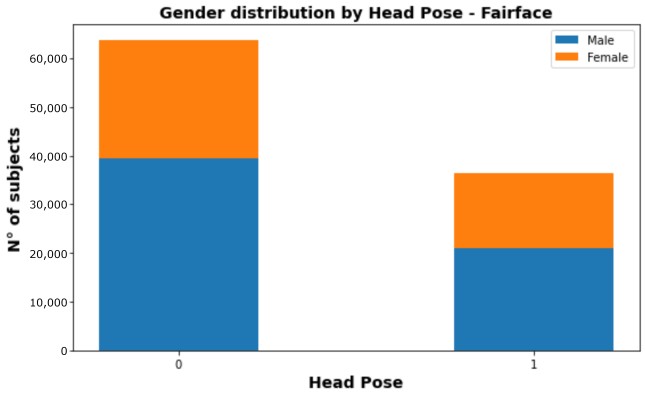

**Figure 3.** HEAD_POSE distribution of the Fairface dataset.

In Step (2), this model achieved an accuracy of 89% during training. Table 2 shows the model performance, and Figures 3 and 4 highlight the AUPRC and specificity metrics, both of which have values of 0.89 and are relevant because the dataset is imbalanced.

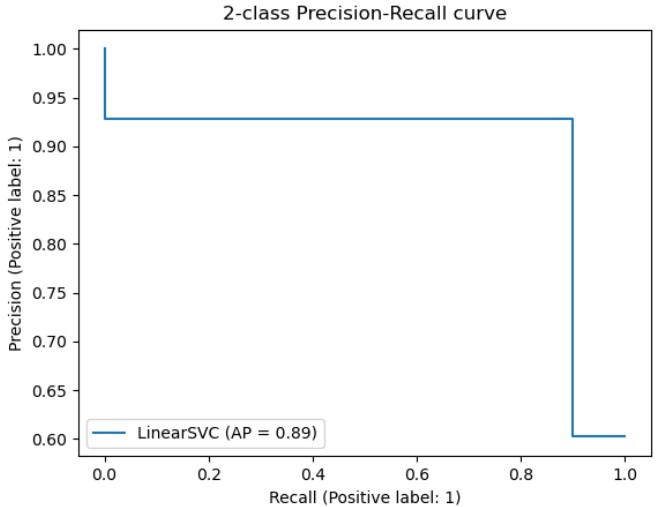

**Figure 4.** AUPRC for VGG19.

**Table 2.** Classification Report VGG19.

|  | Precision | Recall | F1-Score |
|---|---|---|---|
| 0—frontal position | 0.85 | 0.89 | 0.87 |
| 1—another position | 0.92 | 0.89 | 0.91 |
| macro avg | 0.89 | 0.89 | 0.89 |
| weighted avg | 0.89 | 0.89 | 0.89 |

*3.2. Analysis of the Developed Model for the JUBTC Problem*

In Step (1), the toxicity value of the comments was categorized as a target, with 0 being non-toxic and 1 being toxic. Furthermore, Table 3 depicts the discretized data regarding gender in order to facilitate analysis of bias and fairness. Genders are represented by values ranging from 0 to 3, while the absence of this attribute is represented by a value of $-1$.

**Table 3.** Gender categories for Jigsaw.

| Gender | Label |
|---|---|
| Male | 0 |
| Female | 1 |
| Transgender | 2 |
| Other | 3 |

Figure 5 illustrates the imbalance of the dataset in terms of the toxicity of the messages and their corresponding gender. The *other gender* class has few samples that are indistinguishable in the graph.

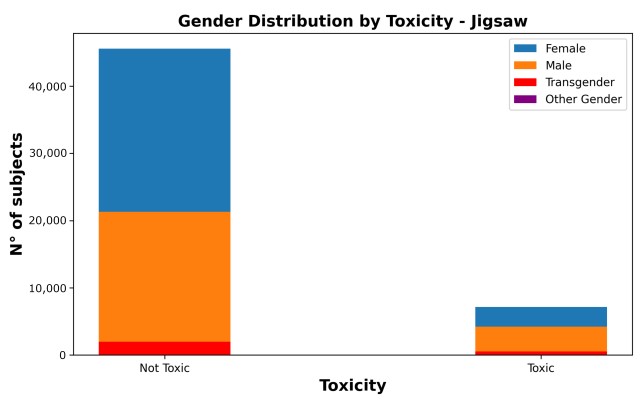

**Figure 5.** Dataset distribution for JUBTC.

In Step (2), the BERT model was evaluated using the accuracy, recall, F1-score, and precision, and the accuracy was 98%, as shown in Table 4. Because the dataset is imbalanced, as shown in Figure 5, the AUPRC seen in Figure 6 is 0.76 and the specificity metric is 0.98.

**Table 4.** Classification Report BERT TensorFlow.

| | Precision | Recall | F1-Score |
|---|---|---|---|
| 0—non-toxic | 0.99 | 0.99 | 0.99 |
| 1—toxic | 0.85 | 0.89 | 0.87 |
| macro avg | 0.92 | 0.94 | 0.93 |
| weighted avg | 0.98 | 0.98 | 0.98 |

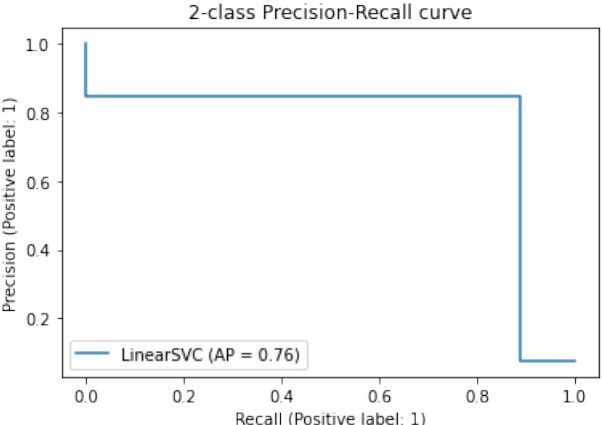

**Figure 6.** AUPRC metric BERT.

### 3.3. Analysis of the Developed Model for the MovieLens Problem

In Step (1), Figure 7 shows how the dataset's gender attribute is imbalanced, with the prevalence of the male class (blue bar) over the female class (orange bar). The x-axis shows user-rated movie stars from 1 to 5, with 4 being the most common.

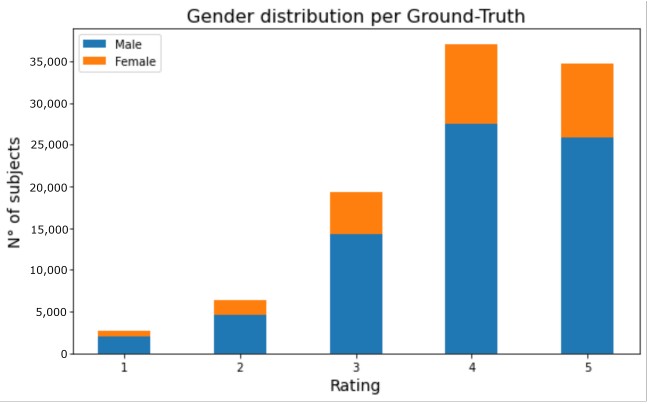

**Figure 7.** MovieLens dataset distribution.

Wide and Deep

In Step (2), the Wide and Deep model developed obtained for the evaluation metrics AUPRC with a value of 0.99 (Figure 8) and accuracy with a value of 0.89 seen in Table 5, as well as specificity with 0.96.

**Table 5.** Classification Report Wide and Deep.

|  | Precision | Recall | F1-Score |
|---|---|---|---|
| 0—not-recommended | 0.44 | 0.96 | 0.61 |
| 1—recommended | 1.00 | 0.88 | 0.93 |
| macro avg | 0.72 | 0.92 | 0.77 |
| weighted avg | 0.95 | 0.89 | 0.90 |

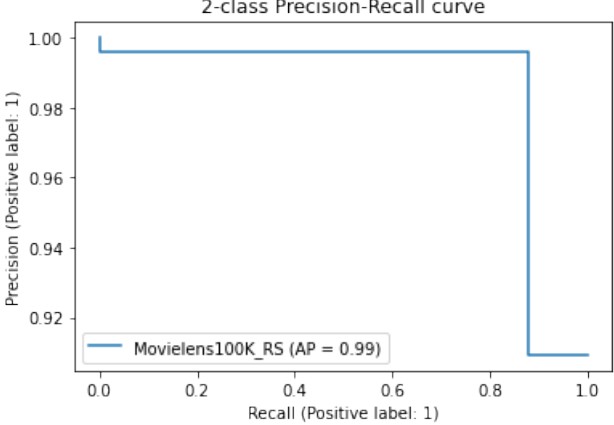

**Figure 8.** AUPRC metric Wide and Deep.

### 3.4. Fairness Metrics in the Models

In Step (3), the fairness metrics for the models were obtained, and these results allowed for the identification of a pattern in Step (4).

In FIFC and MovieLens, those classes are male and female, while in JUBTC, they encompass male, female, transgender, and other. Note that the sensitive attributes are multi-class for JUBTC and binary in FIFC and MovieLens. The benefited group defined

to calculate the fairness metrics, following Equation (6): male for the FIFC and female for JUBTC and MovieLens.

The following metrics were used in the FIFC, JUBTC, and MovieLens datasets to identify bias and fairness: error disparity, FNR disparity, FPR disparity, PPR disparity, TPR disparity, statistical parity, equalized odds, AUPRC disparity, predictive parity, equal of opportunity, predictive equality, FNR difference, and average odd difference.

Figure 9a depicts that the error disparity, PPR disparity, and statistical parity metrics identified bias and fairness, used in the FIFC model, with the beneficiary class being 0-Male and the other fairness metrics failing to do so.

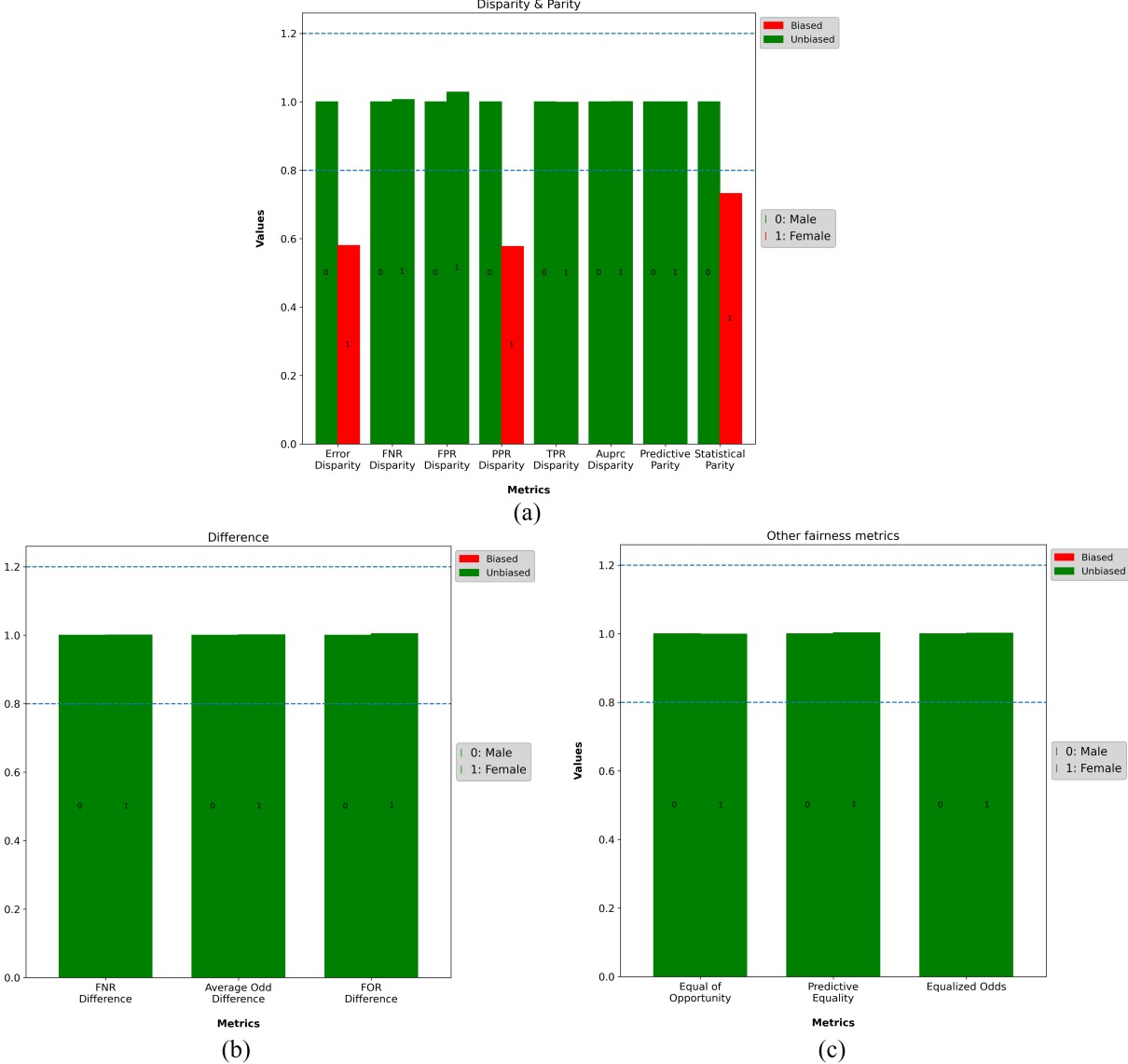

**Figure 9.** VGG19 fairness metrics. (**a**) Results for Disparity and Parity metrics, (**b**) Results for Difference metrics, (**c**) Results for Other Fairness metrics.

Figure 9b,c demonstrates that there was fairness and no bias in the other fairness metrics.

The JUBTC model's fairness metrics were examined using the same method to determine their similarities. Figure 10 illustrates the result of the derived fairness metrics. The occurrence of a gender class in JUBTC, such as male or female, indicates that the message refers to that gender, whether toxic or not. Messages that were not related to gender were removed from the dataset.

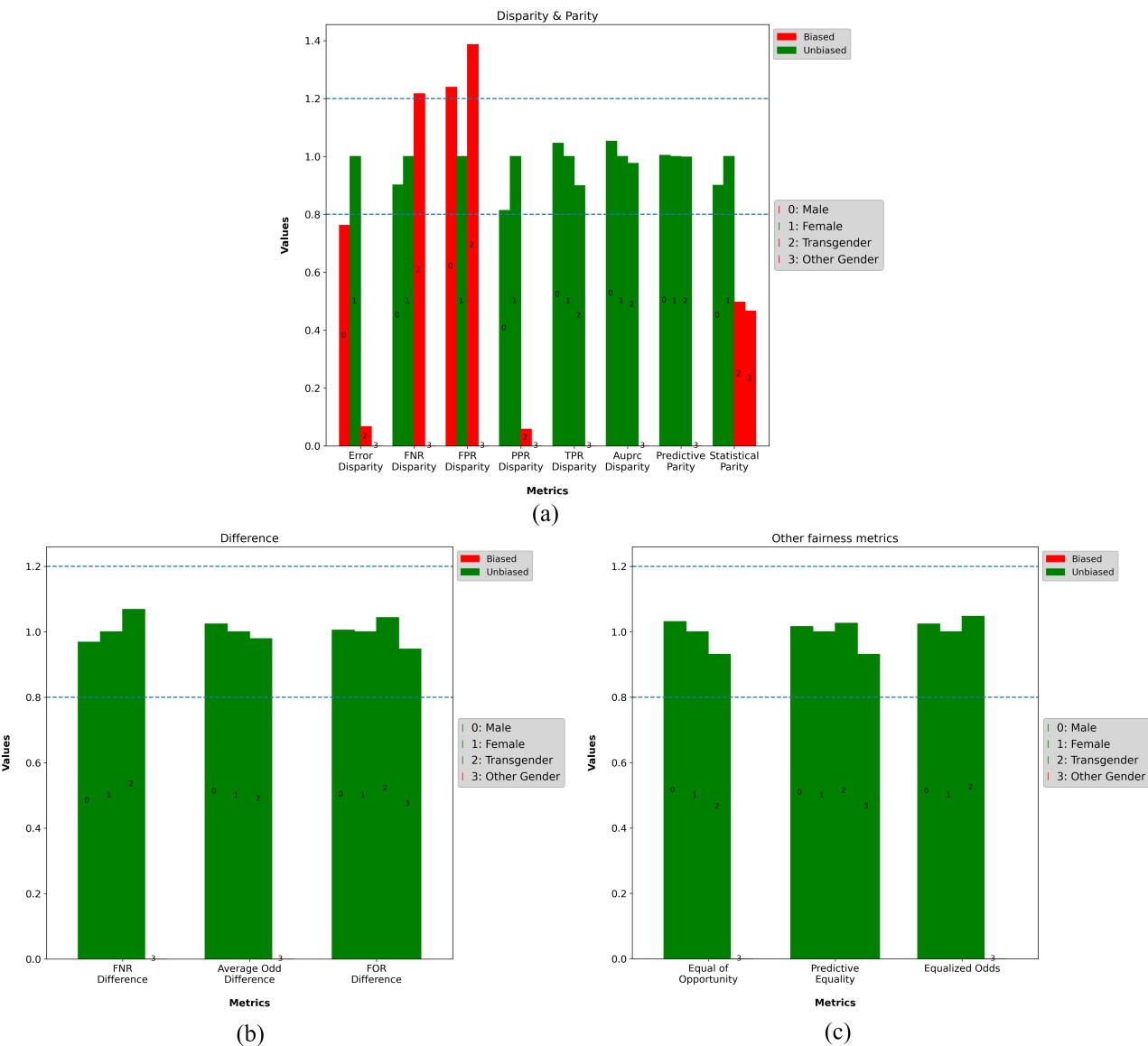

**Figure 10.** BERT fairness metrics. (**a**) Results for Disparity and Parity metrics, (**b**) Results for Difference metrics, (**c**) Results for Other Fairness metrics.

Figure 10a illustrates the disparity metrics. Error disparity, FNR disparity, FPR disparity, and PPR disparity portray that the model is biased and unfair toward male and transgender genders, while in statistical parity, the lack of fairness is identified in transgender and other gender. The TPR disparity, AUPRC disparity and predictive parity portrays fairness and no bias in the model. Due to a lack of data for Group 3, which prevents further analysis, we decided that all metrics for this group should be zeroed out.

Figure 10c depicts that the equalized odds, equal of opportunity, and predictive equality detected no bias and present fairness for male and transgender individuals, neither did the FNR difference, average odd difference and FOR difference (Figure 10b).

For the MovieLens model, the PPR disparity, and error disparity statistical parity metrics in Figure 11a identified bias and lack of fairness, favoring class 1—Male. The other disparity metrics did not indicate bias and indicated fairness, as did the equalized odds metric.

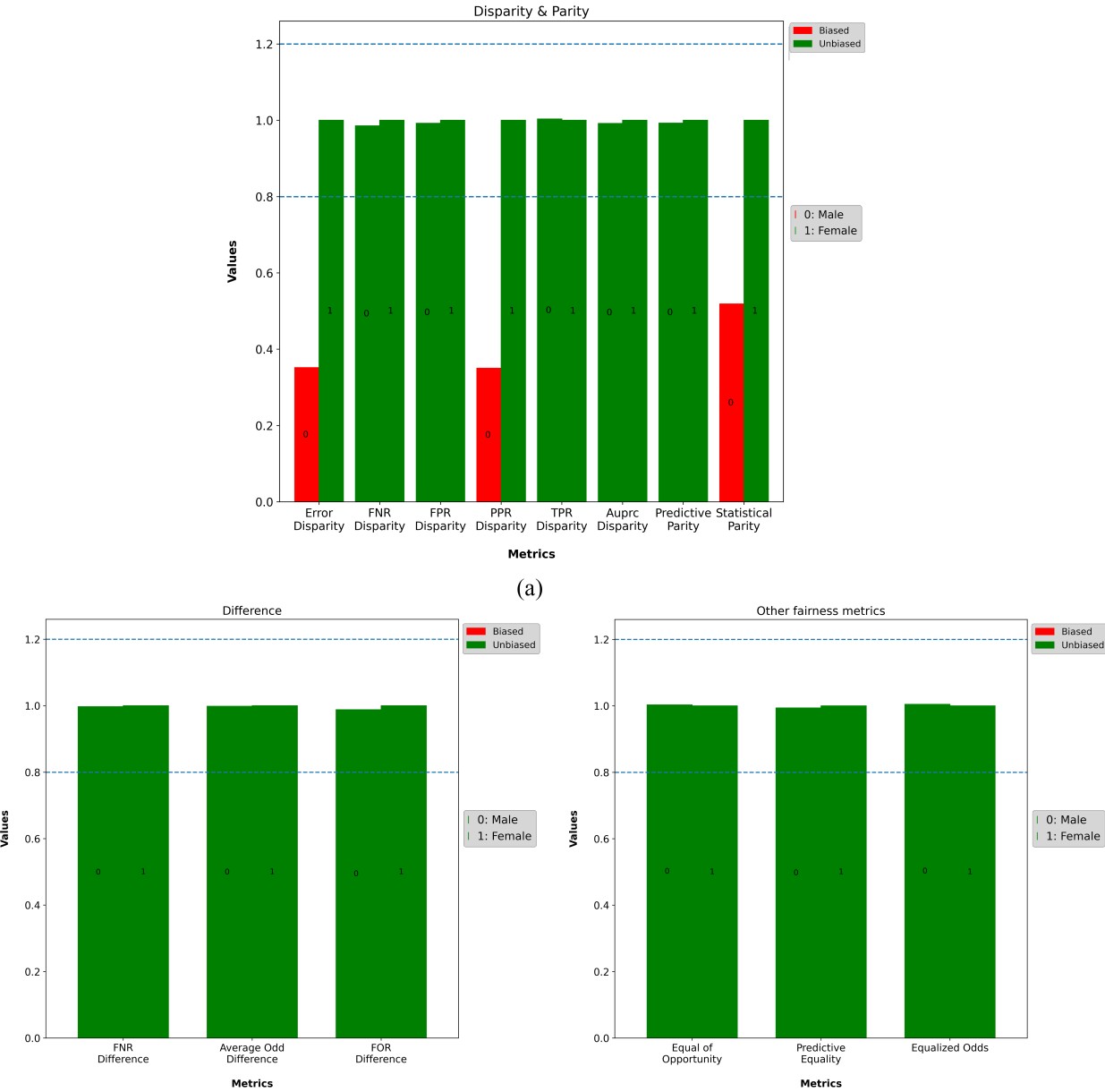

**Figure 11.** Wide and Deep fairness metrics. (**a**) Results for Disparity and Parity metrics, (**b**) Results for Difference metrics, (**c**) Results for Other Fairness metrics.

Figure 11 illustrates that the remaining metrics do not identify bias and present fairness. Tables 6–8 show the results of the analyses of the fairness metrics of the models that resemble each other.

**Table 6.** Disparity and parity metrics in all models.

|  | ED | FNRD | FPRD | PPRD | TPRD | AUPRC | PP | SPV |
|---|---|---|---|---|---|---|---|---|
| FIFC-VGG19 | unfair | fair | fair | unfair | fair | fair | fair | unfair |
| JUBTC-BERT | unfair | unfair | unfair | unfair | fair | fair | fair | unfair |
| MovieLens-Wide and Deep | unfair | fair | fair | unfair | fair | fair | fair | unfair |
|  | Similar | Different | Different | Similar | Similar | Similar | Similar | Similar |

**Table 7.** Difference metrics in all models.

|  | **FNRDif** | **AOD** | **FORD** |
|---|---|---|---|
| FIFC-VGG19 | fair | fair | fair |
| JUBTC-BERT | fair | fair | fair |
| MovieLens-Wide and Deep | fair | fair | fair |
|  | Similar | Similar | Similar |

**Table 8.** Other fairness metrics in all models.

|  | **EO** | **EOO** | **PE** |
|---|---|---|---|
| FIFC-VGG19 | fair | fair | fair |
| JUBTC-BERT | fair | fair | fair |
| MovieLens-Wide and Deep | fair | fair | fair |
|  | Similar | Similar | Similar |

The fairness metrics ED, PPRD, TPRD, FNRDif, AOD, FORD, SPV, EO, AUCPRC, PP, EOO and PE had the same behavior in all models, varying in all as unfair or fair, despite changes for benefited classes between experiments. The fairness metrics FNRD and FPRD had variable behaviors in all models, ranging from unfair or fair.

The results show that neither the model nor the benefiting class affect the relationship between the metric and the sensitive attribute. The JUBTC BERT differs from the other problems for the metrics FNR disparity and FPR disparity, highlighted with yellow color in Table 6. Figure 5 demonstrates a deviation from this pattern, with the harmed class as the least represented. One explanation is that gender is multi-class in JUBTC, whereas in the other datasets, it is binary.

These results reveal that, regardless of the model, the sensitive attribute predicts a similar behavior for the metrics, with the sensitive attribute serving as an indicator of the measure to be employed.

However, additional tests employing correlation and similarity metrics are required to establish the optimal fairness metric for a sensitive feature. Additionally, this method and its results may be helpful in identifying which fairness metrics are more relevant and robust than others for detecting bias and fairness in machine learning models, acting as a steppingstone for additional study to tackle this difficult research subject.

### 4. Conclusions and Future Research

This study provided a steppingstone to analyzing patterns in several metrics for identifying bias and fairness, applying the gender-sensitive attribute as a case study, for three different areas of applications in machine learning models: computer vision, natural language processing, and recommendation systems.

We observed that while some metrics indicated bias and lack of fairness, others did not, revealing a consistent pattern in different application domains for the same sensitive attribute, with similarities across domains for the statistical parity, PPR disparity, and error disparity metrics, thus indicating lack of fairness while accounting for the studied sensitive attribute.

All metrics presented equivalent results in identifying bias and fairness for the studied sensitive attribute, except for FNRD and FPRD. This could be due either to variations within the sensitive attribute between datasets (e.g. different number of classes), or to some specificities related to the detection of false negative/positive instances in the protected and unprotected groups of the sensitive attribute. Nevertheless, this supports our findings since the analyzed patterns point toward the possibility of using common metrics for the same sensitive attribute to identify bias and fairness in machine learning models, for different application domains.

Moreover, we found that the similarities between the metrics and their patterns identified when analyzing the sensitive attribute showed to be a significant factor when determining fairness metrics for different contexts. Hence, our approach can support the analysis of the context to determine the most suitable metrics for recognizing bias and fairness in protected groups, driving the development of more bias-free and fairer machine learning models. We also conclude that different use cases with the same sensitive attributes have similar contexts.

We suggest that future research focuses on comprehending fairness metrics in various settings, including a more diversified number of sensitive attributes. To determine which metric should be applied for each issue, further investigations with a larger number of models, under various architectures, can be conducted as well.

**Author Contributions:** Conceptualization, T.P.P., R.B.L., F.V.N.L., R.M.P., G.A.d.S.G., G.O.R.C., E.L.S.O., I.W. and E.G.S.N.; methodology, T.P.P., I.W. and E.G.S.N.; validation, T.P.P., I.W. and E.G.S.N.; formal analysis, I.W. and E.G.S.N.; investigation, T.P.P., R.B.L., F.V.N.L., R.M.P., G.A.d.S.G., G.O.R.C.; data curation, T.P.P., I.W. and E.G.S.N.; writing—original draft preparation, T.P.P., R.B.L., F.V.N.L., R.M.P., G.A.d.S.G., G.O.R.C., E.L.S.O., I.W. and E.G.S.N.; writing—review and editing, T.P.P., R.B.L., F.V.N.L., R.M.P., G.A.d.S.G., G.O.R.C., E.L.S.O., I.W. and E.G.S.N.; visualization, T.P.P., R.B.L., F.V.N.L., R.M.P., G.A.d.S.G., G.O.R.C., E.L.S.O., I.W. and E.G.S.N.; supervision, T.P.P., E.L.S.O., I.W. and E.G.S.N.; project administration, I.W. and E.G.S.N. All authors have read and agreed to the published version of the manuscript.

**Funding:** This publication is the result of a project regulated by the Brazilian Informatics Law (Law No. 8248 of 1991 and subsequent updates) and was developed under the HP 052-21 between SENAI CIMATEC and HP Brasil Indústria e Comércio de Equipamentos Eletrônicos Ltda. or Simpress Comércio, Locação e Serviços Ltda.

**Institutional Review Board Statement:** Not applicable.

**Informed Consent Statement:** Not applicable.

**Data Availability Statement:** Not applicable.

**Acknowledgments:** We gratefully acknowledge the support of SENAI CIMATEC AI Reference Center for scientific and technical support and the SENAI CIMATEC Supercomputing Center for Industrial Innovation. The authors would like to thank the financial support from the National Council for Scientific and Technological Development (CNPq). Ingrid Winkler is a CNPq technological development fellow (Proc. 308783/2020-4).

**Conflicts of Interest:** There are no conflict of interest associated with this publication.

## Abbreviations

The following abbreviations are used in this manuscript:

| | |
|---|---|
| ML | machine learning |
| CV | computer vision |
| NLP | natural language processing |
| RS | recommendation systems |
| AUPRC | area under the precision–recall curve |
| P | ratio of positive |
| N | ratio of negative |
| macro avg | macro average |
| weighted avg | weighted average |
| FIFC | FairFace challenge |
| JUBTC | Jigsaw unintended bias in toxicity classification |
| IJB-C | IARPA Janus Benchmark-C |
| CCD | Civil Comments dataset |
| VGG19 | Visual Geometry Group |
| BERT | bidirectional encoder representations from transformers |
| SPV | statistical parity value |
| EO | equalized odds |

| | |
|---|---|
| PE | predictive equality |
| PPV | predictive parity |
| AOD | average odd difference |
| FNRDif | FNR difference |
| FNRD | false negative rate disparity |
| PPRD | predicted positive ratio disparity |
| FPRD | false positive rate disparity |
| ED | error disparity |
| TPRD | true positive rate disparity |
| TP | true positive |
| TN | true negative |
| FP | false positive |
| FN | false negative |
| AOD | average odds difference |
| FORD | FOR difference |

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
