# Peer review of "Context-Based Patterns in Machine Learning Bias and Fairness Metrics: A Sensitive Attributes-Based Approach"

_2504-2289, doi:10.3390/bdcc7010027_

Round 1

Reviewer 1 Report

Authors presented the research work on Recognizing context-based patterns in bias and fairness metrics in machine learning models. The topic seems interesting and after reading the whole manuscript, following observation have been made.

1. The title of the paper is too long and should be refined.

2. The usage of the English language should be improved in the paper as some sentences are too long and should be refined. Try to avoid personal keywords like we, they etc. in the manuscript.

3. The structure and layout of the manuscript should be consistent throughout.

4. The result section of the manuscript needs improvement and can be presented in a better manner. 

5.  Reference section should be strengthened with the recent ones.

Author Response

Dear reviewer,

Thank you for your valuable comments and suggestions. The responses to the comments are in the attached pdf.

Reviewer 2 Report

The paper is well-written and organised. But I prefer to consider this paper as a review paper more than the article. In addition, I believe if the authors add the equation for each matric more than the description, such as pages 3,4, Accuracy, Recall, etc.

Also, we need to add a description for the dataset used in the model 

Author Response

(The authors gave the same response as above.)

Reviewer 3 Report

The manuscript analyzed the patterns of several bias and fairness metrics for different application domains of machine learning models based on gender. The manuscript mentions that some metrics indicate bias and unfairness, others do not, which shows a consistent pattern in different application domains for the same sensitive attribute, with similarities across domains for the metrics Statistical Parity, PPR Disparity, and Error Disparity, which indicate unfairness while accounting for the studied sensitive attribute. The authors have studied an interesting topic and such a topic will provide good knowledge and data to the reviewers. Overall, the manuscript is well written. However, before publication, it is suggested to address the following points. The authors may follow these suggestions to improve the readability of the manuscript. Some suggestion changes are given below:

a.       Line 23/24:” some attributes did not present such similar pattern” to “some attributes did not present such pattern” or “some attributes did not present similar pattern”.

b.       Line 10: ” For this end” What kind of end? Suggested rephrasing for better understanding.

c.       Line 106:  ” as well as identifying key” to “as well as identify key”.

d.       Line 145: ” to their profession” to “to their professions”.

2.       In my personal opinion, the Abstract and Conclusion should be improved. The length of the abstract and conclusion seems to be more than a formal requirement. For the abstract part, there is too much information given in the abstract. I believe, it should only focus on the summary/abstract of your work and should not provide in-depth details. Moreover, the last line of the abstract seems to be redundant. Additionally, the conclusion part also includes future work. The title should be changed to Conclusion and future work.

3.       To provide better readability to the research community and follow the norms of a formal manuscript, it is suggested to explicitly present a summary of your contribution(s) in the second last paragraph of the introduction. This summary can be in a paragraph or bullet points.

4.       In section 2 i.e., Materials and Methods, it is suggested to explain the steps under headings i.e., Step (1), Step (2), etc. as shown in Figure 1.

5.       The clip art used in Figure 1 may/can mislead some of the readers as they do not seem to conform with the step title, especially (2) and (3). It is suggested to make a formal flowchart and in this flowchart, the authors may use such clip art.

6.       The caption for Figure 1 doesn’t seem to align with the content of the figure. Please reconsider adding an appropriate caption.

7.       It is suggested to make “F1” score consistent throughout the manuscript. For example, on line 114 “f1-score” on line 124 “F1-score” (despite being a heading/start of a sentence, this notation must not change”).

8.       For all references to equations, use round brackets for number references. For example, Equation 1 should be Equation (1).

9.       For explicit Equations use a proper fraction instead of “\” sign. This sign is fine within the main text, but for Equations, it is suggested to use proper fractions. For example, Equation (1) uses “\”. It is suggested to user fraction as used in Equation (2).

10.   There seems to be confusion, so please reconsider the usage of ”unbalanced datasets” i.e., it should be “imbalanced datasets”. Please reconfirm this word and change it accordingly throughout the manuscript. As unbalance refers to a class distribution that was balanced and is now no longer balanced, whereas imbalanced refers to a class distribution that is inherently not balanced.

11.   In section 2.3. For fairness metrics, use a colon or dot sign after each heading i.e., missing ending in (d).

12.   Line 232: What is 1? It is hard to follow, please rephrase for better readability. Please remove the hyperlink from this 1. Perhaps it means: “with values represented as 1”.

13.   What do 0.00 and 1.00 indicate in Table 2?

14.   Moreover, for all figures, it is suggested to convert the font to the font matching the manuscript. (Not necessary and can be ignored).

15.   It is suggested to update Ref [15], and [31]  to peer-reviewed citations. For example, \ref [15] indicates a preprint, however, this paper was later published in the proceeding of a well-reputed conference. Here are bibtex links to consider:

[15] https://dblp.org/rec/conf/emnlp/TenneyWBBCGJPRR20.html?view=bibtex

[31] https://dblp.org/rec/conf/acl/PavlopoulosSDTA20.html?view=bibtex

Author Response

(The authors gave the same response as above.)

Reviewer 4 Report

I like a lot concept of paper as the general studies on the quality of measures are very important. However, the realization of this concept makes a reader lost. The paper, probably due to many authors, is very inconsistent, while its conclusions are doubtful.

1. Definitions at p.3-4 (accuracy, precision, recall, f1-score) should be expressed with TP, TN, FP, FN.

2. It should be well explained what is the accuracy for precision as in table 2. And what are macro avg and weighted avg for precision, recall and f1. This is a very unclear part of the study.

3. In table 3 accuracy, macro avg and weighted avg are always the same – is it a real outcome?

4. paper should define well fairness, unfairness, bias, equality and equality – I got lost

5. Huge problem is with classification on p.6-7 – these are the fairness measures (l.231) and they can be grouped into 4 classes: disparity, parity, difference and fairness. So first, fairness (subgroup) is a part of fairness (supergroup), but also fairness (supergroup) can be sth else than fairness (subgroup) – a total mess. Secondly, what is the difference between parity and disparity

6. Problem linked to classification on p.6-7 are visualizations on p.12-13 – p.6-7 make 4 groups, while graphs are divided into 5 subgroups. Please make consistent

7. table 8 reports 6 fairness measures, but classification p.6-7 has only 4. Please make consistent

8. Tables 6,7 and 8 include symbols E & I for Equality and inequality – no definition of how to get them. Additionally, columns are summarized as EAM or ETM – again, no explanation of what is that.

9. Conclusions are very poor – authors write softly to say nothing and advertise making more studies to get any knowledge out of it (lines 400-404) – this study should be much more conclusive, what should be written very precisely

10. minor – in fig.2 middle box is the same for each model – in the current form reader looks for differences – it should be made in one box

11. what for you introduce representativeness in eq.2 if it is never used?

12. fig.1 is very general and applies to all studies – here it should be much more particular to illustrate real actions in the study

13. fig.9,10,11 show bias – how it was found that in given summary we observe bias – it must be explained. 

Author Response

(The authors gave the same response as above.)

Round 2

Reviewer 4 Report

Thank you for all corrections, paper is suitable for publication